

# Soil solution phosphorus turnover: derivation, interpretation, and insights from a global compilation of isotope exchange kinetic studies

Julian Helfenstein[1], Jannes Jegminat[2], Timothy I. McLaren[1], Emmanuel Frossard[1]

[1]Institute of Agricultural Sciences, ETH Zürich, Lindau, 8315, Switzerland
[2]Institute of Neuroinformatics, University of Zürich and ETH Zürich, Zürich, 8057, Switzerland

*Correspondence to*: Julian Helfenstein (julian.helfenstein@usys.ethz.ch)

**Abstract.** The exchange rate of inorganic phosphorus (P) between the soil solution and solid phase, also known as soil solution P turnover, is essential for describing the kinetics of bioavailable P. While soil solution P turnover ($K_m$) can be determined by tracing radioisotopes in a soil-solution system, few studies have done so. We believe that this is due to a lack of understanding on how to derive $K_m$ from isotopic exchange kinetic (IEK) experiments, a widespread form of radioisotope dilution study. Here, we provide a derivation of calculating $K_m$ using parameters obtained from IEK experiments. We then calculated $K_m$ for 217 soils from published IEK experiments in terrestrial ecosystems, and also for that of 18 long-term P fertilizer field experiments. Analysis of the global compilation dataset revealed a negative relationship between concentrations of soil solution P and $K_m$. Furthermore, $K_m$ buffered isotopically exchangeable P in soils with low concentrations of soil solution P. This finding was supported by an analysis of long-term P fertilizer field experiments, which revealed a negative relationship between $K_m$ and phosphate buffering capacity. Our study thus highlights the potential of $K_m$ for future studies—not only for P, but also for other environmentally-relevant, strongly-sorbing elements with radioisotopes such as zinc, cadmium, nickel, arsenic, or uranium.

## 1 Introduction

As an essential but often limiting nutrient, phosphorus (P) plays a central role in food production and more efficient P management is key to improve food security (Tilman et al., 2002;Syers et al., 2008). Phosphorus limitation in agroecosystems is usually overcome by applying P fertilizers to the soil surface. However, excessive applications of fertilizer P to soil can cause ecological, societal and economic problems. First, leaching or runoff of fertilizer P from agricultural land to aquatic and marine ecosystems contributes to fish die-off and declining water quality (Carpenter et al., 1998). Second, fertilizer P is largely derived from rock phosphate, which is a non-renewable resource and major deposits are located in only a few countries (Elser and Bennett, 2011;Obersteiner et al., 2013). Third, applications of P fertilizers to soils with a high P sorption capacity can be inefficient because P largely accumulates in the soil as sparingly-soluble forms (Roy et al., 2016).

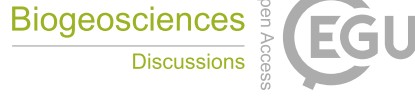



To improve food security while reducing ecosystem pollution, it is essential that we improve our understanding of soil P dynamics, particularly the mechanisms controlling P movement between the soil solid phase and the soil solution where it is bioavailable.

Plants take up P from the soil solution as ionic orthophosphate ($H_2PO_4^-$ or $HPO_4^{2-}$) via roots or mycorrhizal hyphae

(Pierzynski and McDowell, 2005). However, concentrations of P in the soil solution are usually in the order of $10^{-2}$ to $10^1$ mg P $L^{-1}$ (Brédoire et al., 2016), and thus comprise only a small proportion of the total P requirements of plants during a growing season (Pierzynski and McDowell, 2005). Therefore, P exchange kinetics, or the rate at which the soil solution is replenished by P from the soil solid phase, have important implications for the P requirements of living organisms (Menezes-Blackburn et al., 2016;Fardeau et al., 1991). In this study, we investigate a potential link between two different concepts: phosphorus

buffering capacity and isotope exchange kinetics, by analyzing a dataset of global soils and P fertilizer experiments.

Phosphorus buffering capacity (PBC) is defined as the ability of soil to moderate changes in the concentration of soil solution P (Pypers et al., 2006;Olsen and Khasawneh, 1980;Beckett and White, 1964). Historically, PBC has been calculated using Equation 1.

$$PBC = \frac{\Delta \ conc. \ of \ P \ in \ soil \ solution}{\Delta \ conc. \ of \ P \ in \ the \ soil}, \tag{1}$$

The traditional approach of determining PBC of soil is by adding various amounts of P to a soil suspension, equilibrating, and then measuring the slope between sorbed P and P in soil solution. Alternatively, PBC can be measured by analyzing the change in soil solution P concentration with regard to P budget in field P fertilization experiments (Morel et al., 2000). These approaches have revealed that PBC is influenced by ambient temperature, soil solution pH, and concentrations of P in the soil solution, and is highly variable among soil types (Barrow, 1983). One of the most important factors among soil types is

the specific surface area of Fe/Al oxides and clay minerals, which are important sites of P sorption (Gérard, 2016). Whilst the aforementioned approaches are a useful and cost effective way to study soil P dynamics (Bolland and Allen, 2003;Burkitt et al., 2002;Barrow and Debnath, 2014), they are not able to directly determine the turnover of P in the solution.

Isotopic exchange kinetic (IEK) experiments involve the use of P radioisotopes ($^{32}$P or $^{33}$P) to directly measure the exchange of P between the soil solid and solution phases (Frossard et al., 2011). Measures of isotopically exchangeable P are

a more accurate indicator of P bioavailability than that of conventional soil tests based on chemical extraction because the former involve a P radiotracer that can be directly measured and distinguished from all other P ions in the soil (Demaria et al., 2005;Hamon et al., 2002). Previous studies have shown that isotopically exchangeable P is the predominant source of P for most crops (Frossard et al., 1994;Morel and Plenchette, 1994). Though the IEK approach does not consider root-induced pH alterations or secretion of organic acids, increased P availability due to root exudates can be quantified by comparing

isotopically exchangeable P with radioisotope uptake in plants (Hedley et al., 1982). Isotopic dilution in a soil-solution system is characterized by two statistically fitted parameters, *m* and *n* (Fardeau, 1985). The importance of these parameters and their relation to physicochemical properties in the soil was recently investigated (Achat et al., 2016). Furthermore, the





IEK parameters *m* and *n* can be used to calculate the mean soil solution P turnover of a soil, which has been termed $K_m$ (Fardeau et al., 1991;Fardeau, 1985).

Despite several decades of using radioisotopes in P research and the potential relevance of soil solution P turnover to understanding agricultural and natural ecosystems, only six studies have published $K_m$ values, and there has been no
synthesis of these values (Frossard et al., 2011;Fardeau et al., 1991;Fardeau, 1985, 1993;Oberson et al., 1993;Xiong et al., 2002). We believe that this is because an intuitive derivation of $K_m$ has never been published. Whilst information on soil solution P turnover remains limited, $K_m$ values can easily be calculated using data from previously published IEK experiments.

The first aim of our study was to provide a clear and intuitive derivation of the $K_m$ term. Our second aim was to
calculate $K_m$ values from previously published IEK studies, which resulted in a global dataset of over 200 soils. We then tested specific hypotheses related to concentrations of soil solution P and isotopically exchangeable P. Our third aim was to understand the relationship between PBC and $K_m$. This involved an additional dataset based on long-term P fertilizer field experiments, which have reported IEK results and the fertilizer P budgets. Lastly, we carried out a sensitivity analysis of $K_m$ in order to assist interpretation of future results.

Our first hypothesis was that turnover of soil solution P would differ based on soil group. More specifically, we hypothesized that soil groups known to have higher concentrations of sorption sites (such as Andosols and Ferralsols) would have faster turnover rates. Our second hypothesis was that soils with higher concentrations of soil solution P would have lower values of $K_m$ compared to soil with lower concentrations of soil solution P. This is because a high concentration of sorption sites leads to fast absorption and consequently low concentration of P in the solution. Lastly, we hypothesized that
isotopically exchangeable P would be a function of the concentration of P in the soil solution and $K_m$.

## 2. Materials and methods

### 2.1 Derivation of $K_m$

A given volume of soil contains inorganic P in one of two states: the soil phase or the soil solution phase. In any given time interval, physicochemical reactions transfer a fraction of P from the soil solution phase into the solid phase. The rate constant
of this reaction is solution P turnover $K_m$ [min$^{-1}$]. Thus, $K_m$ plays a critical role in determining the time and amount of P available to plants. At low values of $K_m$, there is little exchange.

In equilibrium, an underlying assumption of an IEK experiment, the net flux between the phases is zero because of the balancing effect of the inverse flux, i.e. the flux from the soil phase to the solution phase through desorption and dissolution. In other words, the inverse flux prevents us from measuring $K_m$ directly by fitting the temporal loss of P in soil
solution. If radioisotopes (for P, either $^{32}$P or $^{33}$P) are injected in the soil solution, it becomes possible to experimentally eliminate the inverse flux. Shortly after the injection, the radioisotope is not present in the solid phase and, consequently,



there is no inverse flux. Equation 2 has been found to describe the resulting decline of radioisotope in solution well and is widely used (Fardeau et al., 1991;Frossard et al., 2011).

$$\frac{r_{(t)}}{R} = m \left( t + m^{\frac{1}{n}} \right)^{-n} + \frac{r_{(\infty)}}{R} \tag{2}$$

where $r_{(t)}$ is the radioactivity [Bq] measured at time $t$ [min], $R$ is the total amount of radioactivity added, and $m$ and $n$ are the

model parameters that describe the rapid and slow physico-chemical processes respectively. Since $K_m$ is equivalent to the decline rate of the radioisotope in the absence of an inverse flux, we analyze Eq. (2) right after the injection (t = 0) and find (for details on the derivation, please see Supporting Information):

$$K_{\mathrm{m}} = \frac{n}{m^{\frac{1}{n}}} \tag{3}$$

$K_m$ is thus calculated in three steps: first, $r_{(t)}/R$ is measured, then n and m are determined by non-linear regression, and finally

equation 3 is applied. A limitation of $K_m$ is that it does not take into account an indefinite number of P species each with their own exchange rate (Andersson et al., 2016;Menezes-Blackburn et al., 2016;Gérard, 2016). Also, the IEK method as described above does not consider mineralization of organic P (Oehl et al., 2001). Therefore, the variable $K_m$ should be considered as the average P exchange rate of the soil solution with an indefinite number of solid inorganic P pools.

## 2.2 Dataset

We carried out a literature search for IEK studies reporting $m$, $n$, and $P_w$ values based on the methodological approach of Fardeau et al. (1991). Only values from topsoil layers (0 – 30 cm layer, if reported) were compiled. The dataset includes all papers cited by Achat et al. (2016) in accordance with our aforementioned selection criteria, plus more recent publications. In addition, data obtained from the published literature were supplemented with unpublished data (7 soils), from studies carried out in the Group of Plant Nutrition (ETH Zurich). This resulted in a final dataset of 217 soils taken from 41

references (see Supporting Information Table S1). The soils represented 19 soil groups across the world reference base (WRB, 2015), 26 countries, and all continents except Antarctica. To avoid overrepresentation, sample sizes of two articles reporting many samples of similar soils were randomly reduced, from 30 to 10 (Compaoré et al., 2003) and from 48 to 12 (Tran et al., 1988).

In addition, we carried out a literature search for IEK studies on long-term P fertilizer field experiments. We found

published data across 18 long-term experiment sites (Oberson et al., 1993;Oberson et al., 1999;Fardeau et al., 1991;Gallet et al., 2003;Morel et al., 1994). The soils represented the following soil groups (WRB, 2015): Cambisols, Chernozems, Ferralsols, Fluvisols, Gleysols and Luvisols. In general, the field experiments involved different types of mineral and organic P fertilizers applied at varying rates. The difference in inputs minus outputs led to a range in P budgets from -52 to 125 kg P ha$^{-1}$ yr$^{-1}$.





### 2.3 Data analysis

Isotopically exchangeable P (i.e. E-values: $E_{(t)}$ [mg kg$^{-1}$]), the amount of P that can reach the soil solution within a given time frame is calculated using Equation 4 (Hamon et al., 2002;Fardeau, 1996).

$$E_{(t)} = P_w * \frac{R}{r_{(t)}} \qquad (4)$$

While IEK experiments only last several minutes, $E_{(t)}$ values can be extrapolated beyond the IEK experiment based on Equations 2 and 4 (Frossard et al., 1994;Morel and Plenchette, 1994;Bühler et al., 2003). Extrapolated $E_{(t)}$ values are highly influenced by concentrations of $P_w$. One of the main challenges of the IEK experiment is an accurate and precise determination of $P_w$, particularly in high P-fixing soils (Randriamanantsoa et al., 2013). Analysis involving $E_{(t)}$ could only be performed for studies that reported $P_{inorg}$ in addition to $P_w$, $m$ and $n$.

To examine the relationship between $K_m$ and isotopically exchangeable P, $E_{(t)}$ was calculated for $t = 0$ to 129,600 min using Equation 5 (129,600 minutes is equal to 3 months). First, we calculated the difference between $E_{(1)}$ and $E_{(0)}$ as $\log_{10}(E_{(1)}) - \log_{10}(E_{(0)})$. We then tested if $K_m$ was a significant predictor of this difference using linear regression. To determine the timespan over which $K_m$ affected $E_{(t)}$, we performed linear regression between $K_m$ and $E_{(t)}$ at $t = 1$ to 129,600 min. We also carried out linear regression with $P_w$ and $P_{inorg}$ as predictors of $E_{(t)}$ over the aforementioned time points, respectively. During data analysis, we noticed that different $P_w$ levels were differently sensitive to predictor variables. Therefore, we used Jenks natural breaks optimization to systematically partition the $P_w$ data into three clusters (Bivand et al., 2015).

The sensitivity of $K_m$ to the parameters $m$ and $n$ was calculated using the partial derivatives approach for error propagation (Eq. (5)) (Ku, 1966). By assuming independent errors of the two fitted parameters, we obtain an upper bound on
the error of $K_m$ (Weiss et al., 2006):

$$s_{K_m} = \sqrt{\left(\frac{\partial K_m}{\partial m}\right)^2 s_m^2 + \left(\frac{\partial K_m}{\partial n}\right)^2 s_n^2} \qquad (5)$$

All statistical analyses and graphics were carried out using R (R Core Team, 2017). All model regressions were checked and the model fit determined using significance of fit ($p = 0.05$) and the regression coefficient ($R^2$).

### 2.4 Analysis of long-term field experiments

The fertilizer P budgets were calculated as the average annual input of fertilizer P minus that of crop offtake (kg P ha$^{-1}$ yr$^{-1}$). Each site had three to four P treatments, usually one with a negative budget, one with a balanced budget, and one with a positive budget. To determine the effect of P budget on $P_w$ and $K_m$, we calculated the slope of linear regressions between P budget and $P_w$. The slope of the line relating $P_w$ to P budget can be taken as a field PBC, since the slope of $P_w$ corresponds to the change in $P_w$ over the change in soil P conc. (Eq. (1)). Next, we investigated if there was a relationship between the thus
determined PBC and $K_m$.





## 3. Results and discussion

### 3.1 Global analysis of P turnover in the soil solution ($K_m$)

The turnover rate of P ranged nine orders of magnitude from $10^{-2}$ to $10^6$ min$^{-1}$ across the 217 soils surveyed (Fig. 1).
However, approximately half of the soils had a P turnover rate within the range of $10^0$ and $10^2$ min$^{-1}$. Clear differences in $K_m$

between different soil groups suggest that $K_m$ is related to soil properties governing kinetics of inorganic P in the soil-
solution system. Surface soil horizons of Ferralsols had the highest values of $K_m$, followed by Andosols and Cambisols (Fig.
1). High $K_m$ values of Ferralsols suggest that P in these soils is rapidly adsorbed, i.e. that these soils have a high P buffering
capacity. The lowest $K_m$ values were found in Podzols, which are known to have low P-sorbing capacity (Achat et al., 2009).

Fardeau, Morel, and Boniface (Fardeau et al., 1991) showed that $K_m$ is largest for small values of $n$ and $m$, and

becomes smaller as $n$ approaches 0.5, and $m$ approaches 1, respectively. Values of $n$ and $m$ have often been found to
correlate with soil properties (pH, carbonate concentration, oxalate-extractable Al-/Fe, organic matter, etc.) (Tran et al.,
1988;Demaria et al., 2013;Frossard et al., 1993;Achat et al., 2013). In a global compilation study, it was shown that low
values of $n$ occur for soils with low concentrations of oxalate-extractable Al and Fe, which are indicative of amorphous Al
and Fe oxides (Achat et al., 2016). In contrast, low values of $m$ tend to occur for soils with a low ratio of organic C to Al and

Fe oxides (Achat et al., 2016). The high $K_m$ values of Ferralsols are due to extremely low $m$ values (Mean = 0.025, SD =
0.012, n = 26), and are consistent with low ratios of organic C to Al and Fe oxides typically reported in these soils
(Randriamanantsoa et al., 2013). The Podzols in the dataset, on the other hand, have distinguishably high $m$ values (Mean =
0.50, SD = 0.43, n = 14), which is coherent with the low Al and Fe oxide content of the upper horizon of Podzols (Achat et
al., 2009).

### 3.2 Relationship between soil solution P turnover ($K_m$) and concentration of soil solution P ($P_w$)

There was a negative correlation between $K_m$ and $P_w$, as shown by Fig. (2) and given by Eq. (6):

$$log10(K_m) = 1.26 - 0.960 * log10(P_w) \tag{6}$$

with $F = 127$, $p < 10^{-15}$ and an $R^2 = 0.37$. The two variables $P_w$ and $K_m$ are important in governing plant available P, because
the former describes the amount of P in solution and the latter describes the rate at which it is exchanged. At $t = 1$ min, the

highest values of $E_{(t)}$ occurred for soils with high values of $K_m$ and $P_w$, whereas the lowest values of $E_{(t)}$ occurred for soils
with low values of $K_m$ and $P_w$ (Fig. S1). The relationship was less clear at $t = 1$ day (Fig S1). However, the trend that lowest
E-values occurred for soils with a low $K_m$ and low $P_w$ is still apparent at $t = 1$ day.

The negative correlation between $K_m$ and $P_w$ confirms our second hypothesis and is in accordance with findings
from other studies using different methodological approaches. For example, it has been observed that sorption is less

pronounced on soils with higher P status, due to more negative surface charge (Barrow and Debnath, 2014). The negative
correlation between $P_w$ and $K_m$ fits with this notion. High $K_m$ values imply the presence of many potential binding sites,
where P may adsorb or precipitate. This leads to a rapid exchange between sorption sites and the soil solution, as solution P




quickly binds to a new site. Consequently, $P_w$ is low. On the other hand, slower turnover rates of $P$ in the soil solution and high $P_w$ occur when P-binding sites are few or saturated.

**3.3 Soil solution P turnover ($K_m$) as a buffer of isotopically exchangeable P ($E_{(t)}$)**

We found that $K_m$ is an important predictor of isotopically exchangeable P at exchange times of less than 1 minute. As $t$ increases, $E_{(t)}$ values diverge from $P_w$ and eventually approach $P_{inorg}$. However, the range of calculated $E_{(t)}$ values decreased with time, particularly when t < 1 min. While $P_w$ values ranged almost 4 orders of magnitude, $E_{(1)}$ values only ranged 3 orders of magnitude. Furthermore, differences in $E$-values between soils of low, middle and high $P_w$ decreased with time (Fig 3a). We found that the difference between $\log_{10}(E_{(1)})$ and $\log_{10}(E_{(0)})$ was strongly correlated with $K_m$ ($F = 615$, $p < 10^{-15}$, and $R^2 = 0.79$). Thus, soils with fast rates of $K_m$ had large increases in $E_{(t)}$ compared to soils with slow rates of $K_m$, which showed little difference in $E_{(t)}$ from $E_{(0)}$ to $E_{(1)}$. Furthermore, soils with the largest increases in $E_{(t)}$ had low concentrations of $P_w$ but high values of $K_m$ (Fig. 3b).

While it is evident that $E_{(t)}$ and $K_m$ would be related since both variables are calculated from the same isotope exchange kinetic parameters, the dependency reveals that many soils with low concentrations of $P_w$ "catch up" to other soils due to extremely high soil solution P turnover rates (Fig. 3b). One can thus interpret that a soil with high $K_m$ has a higher PBC and a majority of P applied as fertilizer will be quickly sorbed. On the other hand, high turnover means that there is a large flux of P ions through the soil solution and phosphate ions in solution are quickly replaced through desorption when plants take up P. If soils with $E_{(1min)}$ value of over 5 mg P $kg^{-1}$ are considered highly P-fertile (Gallet et al., 2003), high P fertility can be found in both soils with high $P_w$ and or soils with low $P_w$ but high $K_m$ (Fig. S1). Soils with low $P_w$ and low $K_m$, such as most Lixisols, also have low E-values. Thus, P "fixing" by soils is reversible and says little about P availability.

**3.4 Timeframe over which $K_m$ buffers isotopically exchangeable P ($E_{(t)}$)**

For what time frame is $E_{(t)}$ dependent on $K_m$? By performing linear regressions between $P_w$, $K_m$, and $P_{inorg}$, respectively, and $E_{(t)}$ for t = 1 min to 3 months, we found that the fits are strongly dependent on $P_w$ class (high, middle, low). Based on Jenks natural breaks optimization, three clusters of $P_w$ were determined: 0.008 - 0.16 mg $kg^{-1}$ ($n = 46$), 0.16 – 1.9 mg $kg^{-1}$ ($n = 94$), and 1.9 – 42.5 mg $kg^{-1}$ ($n = 77$). Calculating the $R^2$ of the regression as a function of time showed that for the class of high $P_w$ soils, $P_w$ explained 60% of variability in $E_{(t)}$ at 1 min (Fig 4a). However, $P_w$ lost power as a predictor of $E_{(t)}$ rapidly, explaining only 20% of variability by t = 60 min. In contrast, soils with low concentrations of $P_w$ showed no relationship between values of $E_{(t)}$ and $P_w$ even at low time spans. Thus, the concentration of P in the soil solution has a strong legacy on plant P-availability for soils with high $P_w$ at short time spans, but does not indicate P availability in soils with low concentrations of $P_w$. In these soils, values of $E_{(t)}$ are primarily driven by $K_m$ (Fig 4b). Eventually both $K_m$ and $P_w$ loose predictive power, as $E_{(t)}$ inevitably approaches $P_{inorg}$ (see Eq. (4)) (Fig 4c). However, predictive power of $P_{inorg}$ is again dependent on $P_w$ -class.



Biogeosciences  Discussions

$E_{(t)}$ over timespans between 1 min and 3 months were differently related to predictors $P_w$, $K_m$, and $P_{inorg}$ depending on concentrations of $P_w$. The effect of $K_m$ on $E_{(t)}$ is thus strongly dependent on $P_w$. In P depleted soils the kinetic component is crucial in predicating a soil's P availability. An underestimation of the kinetic components of P availability will lead to overfertilization of P-fixing soils. In more P-rich soils, however, P availability can be relatively accurately assessed with

static measures, i.e. the concentration of P in the solution and the total inorganic P in the soil.

**3.5 $K_m$ buffers fertilizer application in long-term fertilizer experiments**

There was a positive relationship between $P_w$ and P budget across all 18 long-time P fertilizer experimental sites, which is consistent with the study of Morel et al. (Morel et al., 2000). However, the slopes spanned three orders of magnitude, from 0.007 [mg P kg$^{-1}$ soil]/[kg P ha$^{-1}$ yr$^{-1}$] (Ferralsol, Colombia) (Oberson et al., 1999) to 3.9 [mg P kg$^{-1}$ soil]/[kg P ha$^{-1}$ yr$^{-1}$]

(Chernozem, Canada) (Morel et al., 1994). This shows that soil solution P is more buffered in some soils than others. Results from the fertilizer experiments thus confirm that in high P-sorbing soils, such as Ferralsols, additions of P fertilizers may lead to only incremental increases in solution P concentration (Roy et al., 2016). However, this does not necessarily translate to P availability (Pypers et al., 2006).

PBC on the field experiments, taken as the slope of $P_w$ increase with increasing P budget, was negatively dependent

on $K_m$ ($F = 10.8$, $p = 0.0047$, and $R^2 = 0.40$) (Fig. 5). In other words, soils with higher $K_m$ values were characterized by slower increases in $P_w$ at similar yearly P input-output budgets, and vice-versa. Both PBC and $K_m$ are measures to describe the exchange of P between the soil solution and solid phases (Olsen and Khasawneh, 1980;Fardeau et al., 1991). However, the two have never been directly related. Data from long-term field experiments enabled us to compare $K_m$ to field-scale PBC. The fact that the two are correlated in fertilizer field experiments thus underlines our findings from the global soil

investigation that $K_m$ and PBC provide information on the same underlying processes.

**3.6 Implications for using $K_m$**

Most previous studies involving isotopic exchange kinetics have focused on analyzing $m$, $n$, and $E$-values (Achat et al., 2009;Frossard et al., 1993;Tran et al., 1988;Brédoire et al., 2016). However, $m$ and $n$ are simply statistical parameters, whereas $K_m$ is a process-based variable (Fardeau et al., 1991). $K_m$ is the mechanism behind PBC and is useful in explaining P

availability. However, when using $K_m$, it is important to be aware of its limitations (as described in the methods section and in the SI) and $K_m$'s sensitivity to the parameters $m$ and $n$. (Fig. 6) Relatively large errors of up to 100% may be acceptable because differences in $K_m$ between soils or treatments often vary in orders of magnitude (Frossard et al., 2011;Fardeau et al., 1991). However, for low $m$ and or low $n$ values, $K_m$ calculation becomes very sensitive to uncertainty in $m$ and or $n$, and relative errors may be much higher than 100% (Fig. 6). Future studies should take this into account and conduct appropriate

error propagation, or consult Fig. 6 to get an overview of sensitive $m$ and $n$ ranges.

While we focused our analysis on P studies, the derivation of $K_m$ as well as the finding that there is extremely rapid exchange between solid and liquid phases is equally relevant for other nutrients and or pollutants with strongly-sorbing ion



species. The isotope exchange kinetic approach has also been successfully applied to study availability of Zn (Sinaj et al., 1999), Cd (Gray et al., 2004;Gérard et al., 2000), Ni (Echevarria et al., 1998), As (Rahman et al., 2017), and U (Clark et al., 2011), and applications are also plausible for other elements with radioisotopes. Isotope exchange kinetic studies with Zn, Cd, and Ni have used the same method as studies on P analyzed here, also modeling the decline in radioactivity using Eq. (2)

(Gray et al., 2004;Sinaj et al., 1999;Echevarria et al., 1998). For such studies, the derivation of $K_m$ as it is presented here is directly transferable and might provide additional useful information for understanding soil-solution exchange.

### 3.7 Environmental implications

Our study provides new insight on the diffusion-based mechanisms of P buffering across a large range of soil types. Prior to this study little was known about soil solution P turnover rate, as previously $K_m$ had been calculated by only a handful of

studies. Our analysis of 217 soils showed that $K_m$ is inversely proportional to $P_w$ and is an important determinant of plant available P. Biological adaptations to P availability have received a lot of attention, as it has been shown that plant communities have different strategies for P nutrition depending on P availability (Lambers et al., 2008). Indeed, biological activity acts as an important buffer of P in many ecosystems, with higher fluxes of biological P often occurring when there are lower fluxes of physicochemical P (Bünemann et al., 2016;Bünemann et al., 2012). Our global compilation of 217

samples demonstrated there is another buffer of soil solution P, which is independent of biological activity and exclusively diffusion-based. Soils with a low concentration of P in the soil solution tend to have a high P turnover rate, thus buffering isotopically exchangeable P values. This does not mean that negative balances of P will improve the availability of soil P for plant uptake, rather it explains why changes in P availability are not as large as suggested by more drastic changes in $P_w$.

Our findings complement the notion that there are two categories of soils in regard to P dynamics. In many low $P_w$

soils, sorption is extremely high and the soil solution is buffered to P inputs or outputs (Barrow and Debnath, 2014). For these soils, the prevalence of sites with fast exchange rates is crucial to assure a steady flux of P to the soil solution (Fig. 3b). In terms of agricultural management, in such a soil, P fertilization has to be higher than P output via crop removal to account for the buffering effect (Roy et al., 2016). However, once a soil reaches a certain P level and binding sites are saturated by phosphate and other anions, P exchange is less important and fertilizer inputs can be lowered to equal crop offtake (Syers et

al., 2008). For these soils, additional P inputs will directly reflect in an increase in P in the soil solution and P availability is largely driven by the amount of P in the soil solution (Fig. 4a). A better understanding of P kinetics in soil will allow more effective nutrient management to meet the dual goals of improving agricultural production while reducing fertilizer use and pollution.

### Data availability

The global soil aswell as the fertilizer field experiments datasets used in this study are available via the supporting online material.





**Information about the supplement**

The figure relating E-values to Pw and Km is available in the supplement.

**Author contributions**

The project was conceived and carried out by JH with support from EF, TM, and JJ. JJ provided the derivation of $K_m$. JH prepared the manuscript with contributions from all co-authors.

**Competing interests**

The authors declare that they have no conflict of interest.

**Acknowledgements**

We thank Dr. Astrid Oberson for her helpful comments. The project was funded by the Swiss National Science Foundation (Project number 200021_ 162422), which is gratefully acknowledged.

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





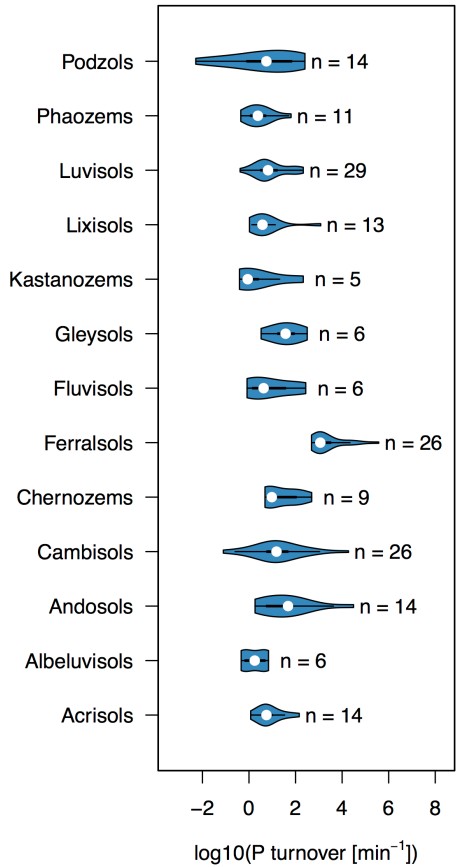

**Figure 1: Violin plots of P turnover ($K_m$) for different world reference base soil groups. Only soil groups with at least five observations were plotted. The number of observations in each violin is written next to the plot. Violin plots are a combination of boxplots and kernel density plots; the point represents the median value and the outline represents the probability density distribution (Adler, 2005).**





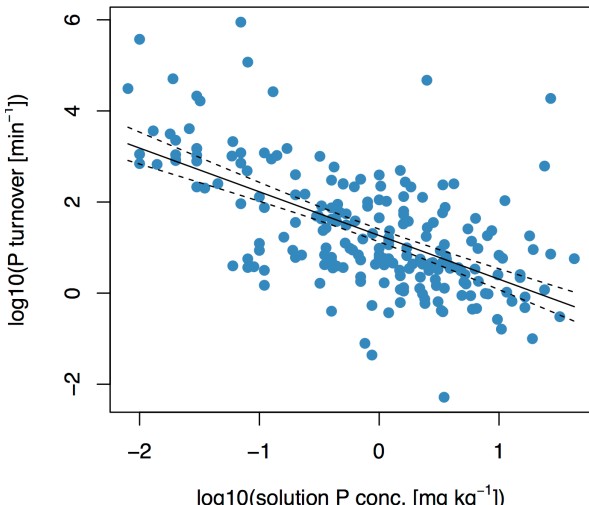

**Figure 2: Simple linear regression between soil solution P turnover ($K_m$) and soil solution P concentration ($P_w$) for 217 soils. The equation is given by $log10(K_m) = 1.26 - 0.960 * log10(P_w)$ with $F = 127$, $p < 10^{-15}$ and an $R^2 = 0.37$.**





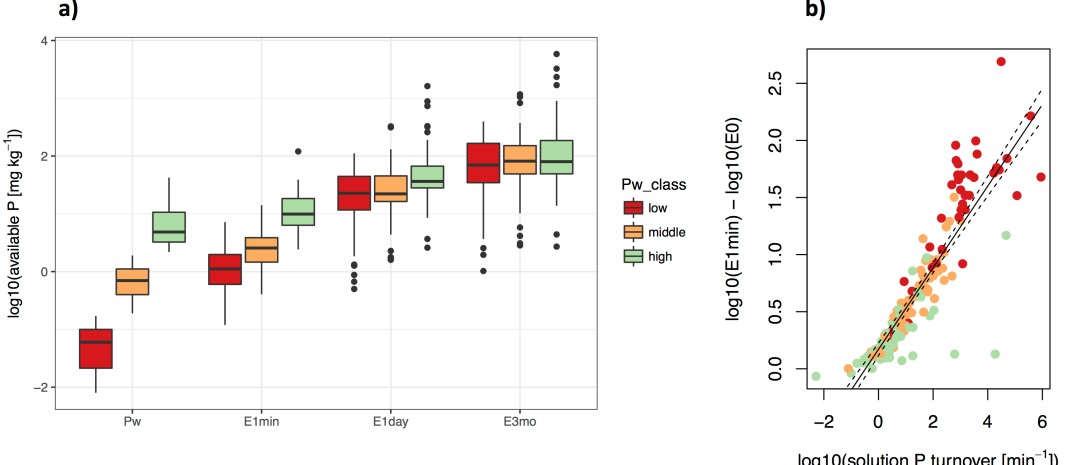

**Figure 3:** Soil solution P turnover ($K_m$) as a driver of available P ($E_{(t)}$). While there is a large range in P availability at t = 0 ($P_w$), this variability becomes smaller and gradually uncoupled from $P_w$ class for longer time frames (t=1, 1440, 129'600 min) (a). The growth in P availability between t=0 and t = 1 is dependent on $K_m$ (b). Simple linear regression between $K_m$ and the difference between $E_{(1)}$ and $E_{(0)}$ is given by $\log10(E_{(1)}) - log10(E_{(0)}) = 0.170 + 0.357 * log10(K_m)$ with F = 615, p < $10^{-15}$, and $R^2$ = 0.79. n = 170. Red, orange, and green colors refer to low, middle, and high $P_w$-classes as determined by Jenks natural breaks optimization.

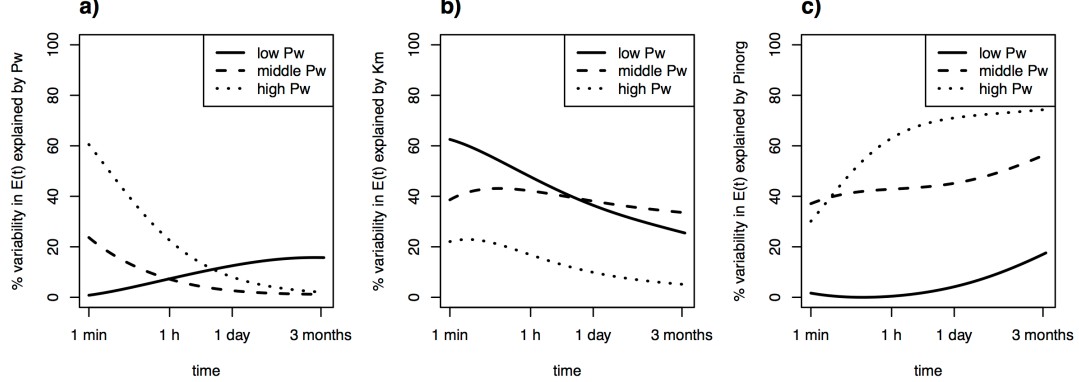

**Abbildung 4:** $R^2$ of simple linear regressions between isotopically exchangeable P ($E_{(t)}$) explained by predictors $P_w$ (a), $K_m$ (b), and $P_{inorg}$ (c) as a function of time. Regressions were fit separately for each class of $P_w$ (low, middle, high), as determined by Jenks natural breaks optimization. Low $P_w$ = 0.008 - 0.16 mg kg$^{-1}$ ($n$ = 46), middle $P_w$ = 0.16 – 1.9 mg kg$^{-1}$ ($n$ = 94), and high $P_w$ = 1.9 – 42.5 mg kg$^{-1}$ ($n$ = 77).





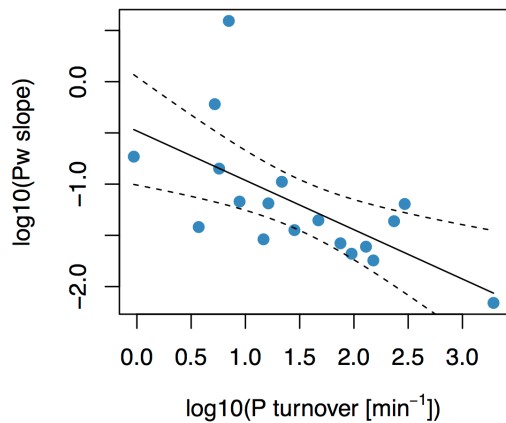

**Figure 5: Simple linear regression between phosphorus buffering capacity (PBC) and soil solution P turnover ($K_m$) for 18 long-term P fertilizer experiments. PBC was calculated as the slope of the regression between $P_w$ and P budget. PBC was found to correlate with $K_m$, as given by, $log10(PBC) = -0.481 - 0.482 * log10(K_m)$, with an $R^2$ of 0.40 ($F = 10.8$, $p = 0.0047$).**





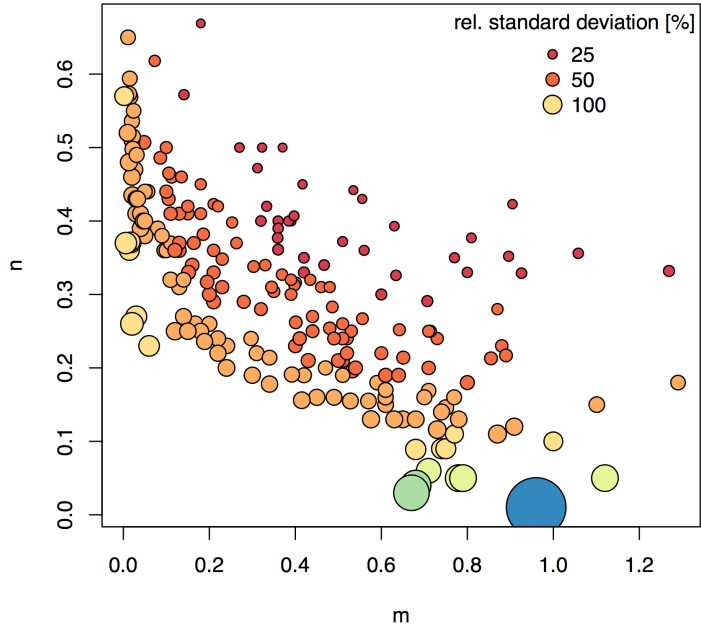

**Figure 6: Sensitivity analysis of $K_m$ to $m$ and $n$ input parameters. The plot shows the $m$ and $n$ values from the 217 soils included in this global compilation study. To show sensitivity of $K_m$, we assumed relative standard deviations (RES, standard deviation/mean [%]) of 10% for each reported $m$ and $n$. Uncertainty was then approximated using the partial derivatives approach. Bubble size and color relates to the RES of $K_m$ for the plotted $m$ and $n$ combination.**