# Peer review of "Soil solution phosphorus turnover: derivation, interpretation, and insights from a global compilation of isotope exchange kinetic studies"

_Biogeosciences, 2017_

## Referee Comment (RC1) · Anonymous Referee #1 · 8 Sep 2017

In general, this paper presents some very interesting results on soil solution phosphorus (P) turnover, which, as the authors pointed out, is a very important concept in describing the kinetics of bioavailable P.

However, I do have several concerns about the methodology and interpretations of results.

The major concern I have is the possible impacts of microbial processes on the results. The authors did not clarify the possible impacts of microbial uptake and turnover in the paper, but emphasizing the new insight is about the diffusion-based mechanism. One guess I have is that the authors accept the assumption from isotopic exchange kinet-

ics studies that during the short-term batch experiment (100 minutes), there is only physiochemical exchange but no biological exchange. It will be better if this argument is clearly stated in the beginning of the method section. Moreover, assuming this assumption is taken for granted, there is still recent evidence showing the strong active role of microbes during the short-term batch experiment (Bunemann et al. 2012). It also seems that the microbial inhibitors don't always work as a perfect solution due to various reasons (Bunemann et al. 2015). It would be not only interesting but also necessary to see if any results of microbial impacts could be drawn from the current dataset.

The second concern I have is about the evolution of the equation 2 and also the determination of parameter m in the dataset. As far as I know, there is a simple version, a version without the $r(\infty)/R$ term, and a full version of the equation from papers in the dataset; and for the parameter m, it is sometimes directly using the value $r(1)/R$ and sometimes a fitted value. How reliable are the results given the huge inconsistency of the dataset, particularly because Km derivates from the full version of the equation and is calculated using m and n values?

The third concern is that some of the hypothesis and discussion section are seemingly self-verifying. For example, in the third hypothesis, E(t) is mathematically already defined as a function of Pw (Eqn. 4), meaning the authors are only looking at E(t) and Km; in section 3.3, the authors concluded that Km is 'an important predictor of isotopically exchangeable P at exchange times of less than 1 minute', but in fact it is because it is defined/derived in this way mathematically (as shown in SI). I would suggest reconsidering some of the sayings used in the paper, as the authors have already mentioned that many of the terms discussed are calculated by the same parameters.

Some technical/specific corrections:

P1, L25-30: the sequence of the three points is a bit difficult to follow

P2, L11: PBC should be abbreviated here rather than at L15

P2, L15: any reference for it?

P2, from L23: from the content of the paper, Km is the main topic, but this is not mentioned in L10 ('In this study, we investigate...'). And it came too late in this paragraph, would be better if it comes earlier and uses an equation, in parallel to PBC.

P3 L5: as far as I know, Frossard et al. 2011 is a book chapter which doesn't publish any new data, maybe cite this in another way?

P5 L29: no need for the abbreviation of conc.

P7 L23: misuse of hyphen

P7 L29: loose (typo)

SI: the numbering and alignment of equations

---

## Author Comment (AC1) · 11 Oct 2017

Here we would like to shortly address the main concern of reviewer one, concerning possible impacts of microbial processes on the results. Indeed, we accepted the basic assumption of the isotope exchange kinetic experiment that the dilution of radioisotopes is purely due to physico-chemical reactions. We think this is a fairly robust assumption since microbial turnover is generally perceived to occur in the timeframe of days to weeks (Oehl et al. 2001), while the turnover controlled by surface reactions occurs within seconds. We need to point that out more clearly in the manuscript and will make sure to do so in the revision.

While it has been shown that in special cases microbial processes affect radioisotope dilution in the short-term batch experiment (Bünemann et al. 2012), this seems to be an exception rather than the rule, and only the case for soils with high microbial activity but low sorption / desorption. Thus, in studies following the 2012 study, the authors wrote that pre-tests showed no impact of a microbial inhibitor and use of a microbial inhibitor during the isotope exchange kinetic experiment was deemed unnecessary (Randria-manantsoa et al. 2015, Bünemann et al. 2016, Wyngaard et al. 2016, Schneider et al. 2017). These studies covered a wide range of land uses, soil characteristics, and geographical locations. The studies support the traditional assumption that only physiochemical processes influence short-term P exchange. Thus, we deduce that while microbial processes may have impact on short-term P exchange in certain soils, this is the exception and not the rule. We maintain that all future studies should test the necessity of a microbial inhibitor. However, we don't think that the results of our meta-analysis are affected by the fact that some (especially earlier) studies did not test the use of microbial inhibitors.

Bünemann, E. K., S. Augstburger, and E. Frossard. 2016. Dominance of either physicochemical or biological phosphorus cycling processes in temperate forest soils of contrasting phosphate availability. Soil Biology and Biochemistry 101:85-95. Bünemann, E. K., a. Oberson, F. Liebisch, F. Keller, K. E. Annaheim, O. Huguenin-Elie, and E. Frossard. 2012. Rapid microbial phosphorus immobilization dominates gross phosphorus fluxes in a grassland soil with low inorganic phosphorus availability. Soil Biology and Biochemistry 51:84-95. Oehl, F., A. Oberson, M. Probst, A. Fliessbach, H.-R. Roth, and E. Frossard. 2001. Kinetics of microbial phosphorus uptake in cultivated soils. Biology and Fertility of Soils 34:31-41. Randriamanantsoa, L., E. Frossard, A. Oberson, and E. K. Bünemann. 2015. Gross organic phosphorus mineralization rates can be assessed in a Ferralsol using an isotopic dilution method. Geoderma 257–258:86-93. Schneider, K. D., R. P. Voroney, D. H. Lynch, A. Oberson, E. Frossard, and E. K. Bünemann. 2017. Microbially-mediated P fluxes in calcareous soils as a function of water-extractable phosphate. Soil Biology and Biochemistry 106:51-60. Wyngaard,

N., M. L. Cabrera, K. A. Jarosch, and E. K. Bünemann. 2016. Phosphorus in the coarse soil fraction is related to soil organic phosphorus mineralization measured by isotopic dilution. Soil Biology and Biochemistry 96:107-118.
* * *

---

## Referee Comment (RC2) · Anonymous Referee #2 · 24 Oct 2017

**General comments**

Helfenstein et al. propose an interesting article about the turnover of P in the soil solution as estimated by isotope exchange kinetics (IEK) experiments, so called $K_m$.

The authors argue that $K_m$ is one of the keys to understand P plant-availability and the underlying mechanisms. They raise the point that, despite its conceptual definition and its derivation were proposed decades ago, this parameter is barely computed and discussed in the IEK literature. To overcome this, they propose a new way of deriving it from the other parameters obtained by IEK experiments. I agree with them that this demonstration is probably more "universally" accessible in the way it does not require

the use of Laplace transforms, as proposed by Fardeau (1996).

Taking advantage of a large compilation of existing IEK data from the upper layer of diverse soils, Helfenstein et al. show that $K_m$ varies among soil types in a way that is coherent with soil properties that are known to influence P dynamics between the solid and the liquid phases of the soil. Together with the concentration of P ions in the soil solution ($P_w$), $K_m$ allows a mechanistic understanding of the value of isotopically exchangeable P ($E_{(t)}$) and, beyond that, the P fertility of a given soil. The authors also show that $K_m$ is rather well correlated with P buffering capacity (PBC) as evaluated on long-term fertilization experiments.

I found particularly appealing the study of the proportion of the variation of $E_{(t)}$ that can be explained by $P_w$, $K_m$, and $P_{inorg}$ (Fig. 4).

Concerning the impact of microbial activity on the results, as raised by referee 1 (see the public discussion), I agree with the response of the authors. This study has to be placed in the framework of IEK experiments with their inherent assumptions.

Globally, I found this manuscript rather clear and concise. The objectives and hypotheses are well stated and relevant—at the exception of the last hypothesis (see below)— and the results are interestingly presented and discussed. The supplementary material is also relevant. I recommend the publication of this study in *Biogeosciences* without major concerns. I provide some specific and technical comments in the next two sections.

**Specific comments**

p. 2, l. 5: "concentrations of P in the soil solution..." this term could be misleading for those who are not familiar with IEK experiments, particularly in the introduction. It could be confused with field measurements while it is the concentration in the conditions of the IEK experiment.

p. 2, l. 5–7: the progression of ideas is not straightforward, what are these "total P

requirements" (provide some examples)? How $P_w$ is related with them?

p. 3, l. 19–20: as formulated, the last hypothesis seems an evidence. In fact, $E_{(t)}$ is a function of $P_w$, and $m$ and $n$ (see Eq. 4 and 2). Please reformulate. Perhaps you wanted to introduce the work presented in Fig. 4. In that case, a suggestion (do what you want with this): "We hypothesized that the dependence of P availability on $K_m$ and $P_w$ evolves with time(, in relation to the different mechanisms involved at different time scales)". Or maybe you wanted to introduce the idea that $P_w$ together with $K_m$ permit to understand P availability (and not $P_w$ or $K_m$ alone)...

p. 4, section 2.2: besides soil types, could you provide some information (such as simple descriptive statistics) on the types of ecosystems (e.g. cropland, pasture, forest, grassland) represented in your dataset?

p. 5, l. 18–21: this MM paragraph on the sensitivity analysis is not clear. Some additional information, such as the assumption of a RES of 10 % for both $m$ and $n$, is provided in the description of Fig. 6 but it should also be provided in the MM. In addition, why to abbreviate "relative standard deviation" as "RES" and not "RSD"?

p. 6, l. 8: "The lowest $K_m$ values were found in Podzols, which are known to have low P-sorbing capacity", however, there is a huge range of $K_m$ values for podzols and the median does not seem to be one of the lowest (Fig. 1). Are there some hypotheses to discuss this? Nevertheless, we approach here the limits of this dataset, which contains only a few values for each soil type—despite being representative of most, if not all, the IEK literature published—and we have no insurance that the median obtained with 5–29 points is truly representative of the soil type.

p. 6, l. 28: remind briefly your second hypothesis.

p. 6, l. 30: what does "P status" mean? Rephrase.

p. 6, l. 30–31: there is no need to repeat what was written two lines before.

p. 7, l. 5–7: "the range of calculated $E_{(t)}$", this is not clear at first read... I suggest to

start l. 6 by "**Indeed,** while $P_w$ values..."

p. 8, l. 26: where in the SI? I did not see it.

p. 8, l. 26: "Relatively large errors...", which errors are you talking about? Rephrase.

p. 8, section 3.6: where do the errors come from? Could something be done to reduce them?

Supplementary material: add the lists of the references used in the two compilation datasets?

**Technical corrections**

p. 5, l. 11: "Eq. 4" instead of "Eq. 5"?

p. 5, l. 16–17 & Fig. 1: it seems you do not cite R packages properly in the text. In fact, it is a more common practice to state in the MM something like "Jenks natural break optimization was performed with the R package 'classInt' v.0.1-24 (Bivand et al, 2015)" right after you wrote you used R for data analyses (p. 5, l. 22). The way you cite Bivand et et (2015) and Adler (2005) seems to refer to the publications where the methods were presented first. Finally, I'm not sure it is useful to provide a citation to justify what is a violin plot or how you performed it.

p. 7, l. 6: do you mean "when t > 100 min" instead of "when t < 100 min"?

p. 7, l. 6: refer here to Fig. 3a

p. 7, l. 8: the linear relation is with $log_{10}(K_m)$, not $K_m$

p. 7, l. 13: "catch up to other soils", rephrase?

p. 8, l. 3: replace "predicating" by "predicting"

p. 8, l. 7: replace "long-time" by "long-term"?

p. 8, l. 8: cite as "Morel et al (2000)"

p. 9, l. 9: add a comma: "Prior to this study**,** little was known..."

p. 9, l. 20: "the soil solution is buffered **by** P inputs"

p. 10, l. 23 & 32: the references for two R packages "Adler (2005)" and "Bivand et al (2015)" look strange, check if no information is missing.

Fig. 2, 3b, and 5: explain what are the black dashed lines (e.g. confidence interval at 95 %).

Fig. 4: is labelled "Abbildung 4"

Fig. 6: add the values higher than 100 % to the legend. Precise in the title of the legend that it concerns the RES of $K_m$. I also suggest to inverse the colour code of the legend (blue/green for small RES and red for high RES). Again, why to abbreviate "relative standard deviation" as "RES" and not "RSD"?

Supplementary information, around the end of p. 1: can we say "concentration of radioactivity"?

---

## Author Comment (AC2) · 3 Nov 2017

In general, this paper presents some very interesting results on soil solution phos- phorus (P) turnover, which, as the authors pointed out, is a very important concept in describing the kinetics of bioavailable P.

*We thank the reviewer for their positive comments.*

However, I do have several concerns about the methodology and interpretations of results.

The major concern I have is the possible impacts of microbial processes on the results. The authors did not clarify the possible impacts of microbial uptake and turnover in the paper, but emphasizing the new insight is about the diffusion-based mechanism. One guess I have is that the authors accept the assumption from isotopic exchange kinetics studies that during the short-term batch experiment (100 minutes), there is only physiochemical exchange but no biological exchange. It will be better if this argument is clearly stated in the beginning of the method section. Moreover, assuming this as- sumption is taken for granted, there is still recent evidence showing the strong active role of microbes during the short-term batch experiment (Bunemann et al. 2012). It also seems that the microbial inhibitors don't always work as a perfect solution due to various reasons (Bunemann et al. 2015). It would be not only interesting but also necessary to see if any results of microbial impacts could be drawn from the current dataset.

*We agree with the reviewer that the role of microbial processes during an IEK experiment needs to be more clearly explained in the manuscript. Please see the 'Interactive comment' published in the online discussion (11/10/2017) for our response related to this comment. We have revised the manuscript to make this clearer (p 2, l 28 and p 4, l 18).*

The second concern I have is about the evolution of the equation 2 and also the deter- mination of parameter m in the dataset. As far as I know, there is a simple version, a version without the r(1)/R term, and a full version of the equation from papers in the dataset; and for the parameter m, it is sometimes directly using the value r(1)/R and sometimes a fitted value. How reliable are the results given the huge inconsistency of the dataset, particularly because Km derivates from the full version of the equation and is calculated using m and n values?

*The reviewer is correct regarding the use of a 'simple' and a 'full' version of equation 2 (see below).*

*Simple version:*  $\quad \dfrac{r(t)}{R} = \dfrac{r(1)}{R} * t^{-n}$

*Full version:*  $\quad \dfrac{r(t)}{R} = m * \left(t + m^{\frac{1}{n}}\right)^{-n} + \dfrac{r(\infty)}{R}$

*From a mathematical point of view, m $= \dfrac{r(1)}{R}$ if $\dfrac{r(\infty)}{R}$ approaches 0 and $m^{\frac{1}{n}}$ also approaches 0. Values derived from these terms tend to be small and differences in the r(1)/R and m parameter using both models are minor in most soils (Fardeau et al. 1991). We have made this clearer in the manuscript (p 4, l 29-31). To make sure that using parameters estimated by the simple model does not bias $K_m$ calculation, we tested this assumption with data from our lab. As shown in Fig. 1, we found that there is no systematic difference between $K_m$ calculated using r(1)/R and n from*

*the simple model or using m and n from the full model. In fact, the difference between the full and the simple model is in the same range as the scatter between replicates of the same soil. We consider this as proof that it is valid to calculate $K_m$ from either parameters estimated by the simple or parameters estimated by the full model.*

[Figure]

Figure 1. Comparison of Km calculated from parameters of the full and simple model for seven different soils, with four replicates each. The line denotes the 1:1 line. These seven soils were chosen because the first author performed IEK analyses on these soils, and thus had the full raw data available to fit both models. "noP" is a Cambisol and the other soils are Andosols with strongly varying P exchange dynamics. For more information on these soils, please see "dataset_soils.xlsx", the supplementary table containing information on all the soils used in the study.

*It should be noted that the effect of the two models on calculated parameters/terms is more pronounced over the long-term, which is particularly the case for E-values due to the missing $\frac{r(\infty)}{R}$ term. In this case, E-values tend to be overestimated using the simple model compared to that of the full model. Therefore, we only used the full model when calculating E-values (Fig. 3*

*and 4) (p 5, l 14-15).*

The third concern is that some of the hypothesis and discussion section are seemingly self-verifying. For example, in the third hypothesis, E(t) is mathematically already de- fined as a function of Pw (Eqn. 4), meaning the authors are only looking at E(t) and Km; in section 3.3, the authors concluded that Km is 'an important predictor of isotopically exchangeable P at exchange times of less than 1 minute', but in fact it is because it is defined/derived in this way mathematically (as shown in SI). I would suggest reconsidering some of the sayings used in the paper, as the authors have already mentioned that many of the terms discussed are calculated by the same parameters.

*In regards to the hypothesis: This concern was also raised by Reviewer 2. We have adapted the suggestion by reviewer 2 (see below).*

*"Lastly, we hypothesized that the dependence of isotopically exchangeable P on $P_w$ and $K_m$ evolves with time."*

*In regards to Section 3.3: Yes, E(t) is mathematically defined as a function of $P_w$ (Eq. 4). In contrast, an analysis of the dataset revealed that $P_w$ has little predictive power for E(t), particularly for soils with low concentrations of $P_w$ (see Fig. 4a). Our results show that $K_m$ is the main driver of P availability at short time spans (Fig. 3b). In, "an important predictor of isotopically exchangeable P at exchange times of less than 1 minute", we changed "predictor" to "buffer".* We were not sure what other sayings the reviewer was concerned about.

Some technical/specific corrections:

P1, L25-30: the sequence of the three points is a bit difficult to follow

*Agreed. We have changed the order of the sentences so that the flow is more logical.*

P2, L11: PBC should be abbreviated here rather than at L15

*Agreed, the term 'Phosphorus buffering capacity' is first used and its abbreviation defined on Page 2, Line 12.*

P2, L15: any reference for it?

*Yes, a reference has been added.*

P2, from L23: from the content of the paper, Km is the main topic, but this is not men- tioned in L10 ('In this study, we investigate. . .'). And it came too late in this paragraph, would be better if it comes earlier and uses an equation, in parallel to PBC.

*Yes, agreed. We have changed the sentence in Line 10 accordingly. Also, we changed the paragraph starting at Line 23 to emphasise the importance of Km in the study, and added the equation for calculating Km as suggested by the reviewer.*

P3 L5: as far as I know, Frossard et al. 2011 is a book chapter which doesn't publish any new

data, maybe cite this in another way?

*The reviewer is correct that this reference relates to a book chapter. The reason it is cited here is because it reports Km values, which are not reported in the original publication of Gallet et al. (2003).*

P5 L29: no need for the abbreviation of conc.

*Corrected.*

P7 L23: misuse of hyphen

*Corrected.*

P7 L29: loose (typo)

*Corrected.*

SI: the numbering and alignment of equations

*Corrected.*

*References*

*Interactive comment on Biogeosciences Discuss., https://doi.org/10.5194/bg-2017-304, 2017.*

*Fardeau, J.-c., C. Morel, and R. Boniface. 1991. Phosphate ion transfer from soil to soil solution: kinetic parameters. Agronomie **11**:787-797.*

---

## Author Comment (AC3) · 3 Nov 2017

Helfenstein et al. propose an interesting article about the turnover of P in the soil solution as estimated by isotope exchange kinetics (IEK) experiments, so called $K_m$.

The authors argue that $K_m$ is one of the keys to understand P plant-availability and the underlying mechanisms. They raise the point that, despite its conceptual definition and its derivation were proposed decades ago, this parameter is barely computed and discussed in the IEK literature. To overcome this, they propose a new way of deriving it from the other parameters obtained by IEK experiments. I agree with them that this demonstration is probably more "universally" accessible in the way it does not require the use of Laplace transforms, as proposed by Fardeau (1996).

Taking advantage of a large compilation of existing IEK data from the upper layer of diverse soils, Helfenstein et al. show that $K_m$ varies among soil types in a way that is coherent with soil properties that are known to influence P dynamics between the solid and the liquid phases of the soil. Together with the concentration of P ions in the soil solution ($P_w$), $K_m$ allows a mechanistic understanding of the value of isotopically exchangeable P ($E_{(t)}$) and, beyond that, the P fertility of a given soil. The authors also show that $K_m$ is rather well correlated with P buffering capacity (PBC) as evaluated on long-term fertilization experiments.

I found particularly appealing the study of the proportion of the variation of $E_{(t)}$ that can be explained by $P_w$, $K_m$, and $P_{inorg}$ (Fig. 4).

Concerning the impact of microbial activity on the results, as raised by referee 1 (see the public discussion), I agree with the response of the authors. This study has to be placed in the framework of IEK experiments with their inherent assumptions.

Globally, I found this manuscript rather clear and concise. The objectives and hypothe- ses are well stated and relevant—at the exception of the last hypothesis (see below)— and the results are interestingly presented and discussed. The supplementary material is also relevant. I recommend the publication of this study in *Biogeosciences* without major concerns. I provide some specific and technical comments in the next two sections.

*We thank the reviewer for their positive comments.*

**Specific comments**

p. 2, l. 5: "concentrations of P in the soil solution..." this term could be misleading for those who are not familiar with IEK experiments, particularly in the introduction. It could be confused with field measurements while it is the concentration in the conditions of the IEK experiment.

*We see the reviewers point. However, we think it would be too specific and would confuse the reader to already talk about IEK experiments in this paragraph. To avoid the confusion, we changed the sentence as follows, "However, concentrations of P in the soil solution are usually small (Brédoire et al., 2016), and in order to meet plant needs P in the soil solution must be replenished continuously (Pierzynski and McDowell, 2005)."*

p. 2, l. 5–7: the progression of ideas is not straightforward, what are these "total P requirements"

(provide some examples)? How $P_w$ is related with them?

*Our aim was to highlight the inability of the P in the soil solution to supply the plant with sufficient P for growth, which would therefore necessitate the resupply of P to the soil solution. See changes to the sentence as written above.*

p. 3, l. 19–20: as formulated, the last hypothesis seems an evidence. In fact, $E_{(t)}$ is a function of $P_w$, and m and n (see Eq. 4 and 2). Please reformulate. Perhaps you wanted to introduce the work presented in Fig. 4. In that case, a suggestion (do what you want with this): "We hypothesized that the dependence of P availability on $K_m$ and $P_w$ evolves with time (, in relation to the different mechanisms involved at different time scales)". Or maybe you wanted to introduce the idea that $P_w$ together with $K_m$ permit to understand P availability (and not $P_w$ or $K_m$ alone)...

*This concern was also raised by Reviewer 1. We agree with the reviewers and have revised the third hypothesis to, "Lastly, we hypothesized that the dependence of isotopically exchangeable P on $P_w$ and $K_m$ evolves with time."*

p. 4, section 2.2: besides soil types, could you provide some information (such as simple descriptive statistics) on the types of ecosystems (e.g. cropland, pasture, forest, grassland) represented in your dataset?

*We have added this information to the manuscript (p 4, l 26-29).*

p. 5, l. 18–21: this MM paragraph on the sensitivity analysis is not clear. Some ad- ditional information, such as the assumption of a RES of 10% for both m and n, is provided in the description of Fig. 6 but it should also be provided in the MM. In addi- tion, why to abbreviate "relative standard deviation" as "RES" and not "RSD"?

We have provided additional information on the sensitivity analysis and made corrections as suggested by the reviewer (p 6, l 23).

p. 6, l. 8: "The lowest $K_m$ values were found in Podzols, which are known to have low P-sorbing capacity", however, there is a huge range of $K_m$ values for podzols and the median does not seem to be one of the lowest (Fig. 1). Are there some hypotheses to discuss this? Nevertheless, we approach here the limits of this dataset, which contains only a few values for each soil type—despite being representative of most, if not all, the IEK literature published—and we have no insurance that the median obtained with 5–29 points is truly representative of the soil type.

*Though the mean is not the lowest, the lowest Km values were from soils belonging to the Podzols group. We agree with the reviewer that with only few samples per soil group one should be cautious to make interpretations about soil groups, which are any way extremely broad and often contain soils whose properties overlap with other soil groups. We made minor changes (p 6, l 15), and added a cautionary sentence, "small sample sizes per soil group and large spans in soil properties even within soil groups mean that group-specific $K_m$ values should not be over-interpreted" (p 6, l 27).*

p. 6, l. 28: remind briefly your second hypothesis.

*Corrected.*

p. 6, l. 30: what does "P status" mean? Rephrase.

*Changed to "heavily fertilized".*

p. 6, l. 30–31: there is no need to repeat what was written two lines before.

*Corrected.*

p. 7, l. 5–7: "the range of calculated $E_{(t)}$", this is not clear at first read... I suggest to start l. 6 by "**Indeed,** while $P_w$ values..."

*We have revised the sentence to make this clearer.*

p.8, l.26: where in the SI? I did not see it.

*In an earlier version of the manuscript we included additional information in the SI, which we later decided to include in the body text of the manuscript. We have removed reference to the SI.*

p. 8, l. 26: "Relatively large errors...", which errors are you talking about? Rephrase.

*We have made this clearer in the manuscript.*

p. 8, section 3.6: where do the errors come from? Could something be done to reduce them?

*As previously identified in methods section 2.3, the errors presented in the sensitivity analysis of this study were calculated assuming relative standard deviations of 10% for the m and n parameters. We did this to highlight the areas in which there is high Km sensitivity, i.e. when m and or n is low. Error propagation is much higher in this area for mathematical reasons. We have made no changes to the manuscript.*

Supplementary material: add the lists of the references used in the two compilation datasets?

*In "dataset_fertilizerexperiments.csv" and "dataset_soils.csv", the references can be found in the 2ⁿᵈ to last and 4ᵗʰ to last column, respectively.*

**Technical corrections**

p. 5, l. 11: "Eq. 4" instead of "Eq. 5"?

*Corrected.*

p. 5, l. 16–17 & Fig. 1: it seems you do not cite R packages properly in the text. In fact, it is a more common practice to state in the MM something like "Jenks natural break optimization was performed with the R package 'classInt' v.0.1-24 (Bivand et al, 2015)" right after you wrote you used R for data analyses (p. 5, l. 22). The way you cite Bivand et et (2015) and Adler (2005) seems to refer to the publications where the methods were presented first. Finally, I'm not sure it is useful to provide a citation to justify what is a violin plot or how you performed it.

*We have made changes to the body text and removed the description of the violin plots.*

p. 7, l. 6: do you mean "when t **>** 100 min" instead of "when t < 100 min"?

*The original sentence was, "However, the range of calculated $E_{(t)}$ values decreased with time, particularly when t < 1 min." We meant that the spread in E(t) was lower at E(1) than E(0) (i.e. Pw). Since this seems to be confusing, we removed this sentence fragment.*

p. 7, l. 6: refer here to Fig. 3a

*Corrected.*

p. 7, l. 8: the linear relation is with $\log_{10}(K_m)$, not $K_m$

*Corrected.*

p. 7, l. 13: "catch up to other soils", rephrase?

*Corrected.*

p. 8, l. 3: replace "predicating" by "predicting"

*Corrected.*

p. 8, l. 7: replace "long-time" by "long-term"?

*Corrected.*

p. 8, l. 8: cite as "Morel et al (2000)"

*Corrected.*

p. 9, l. 9: add a comma: "Prior to this study**,** little was known..."

*Corrected.*

p. 9, l. 20: "the soil solution is buffered **by** P inputs"

*Corrected.*

p. 10, l. 23 & 32: the references for two R packages "Adler (2005)" and "Bivand et al (2015)" look strange, check if no information is missing.

*We have revised the reference to include the package version, which was previously missing.*

Fig. 2, 3b, and 5: explain what are the black dashed lines (e.g. confidence interval at 95 %).

*Corrected.*

Fig. 4: is labelled "Abbildung 4"

*Corrected.*

Fig. 6: add the values higher than 100 % to the legend. Precise in the title of the legend that it concerns the RES of $K_m$. I also suggest to inverse the colour code of the legend (blue/green for small RES and red for high RES). Again, why to abbreviate "relative standard deviation" as "RES" and not "RSD"?

*Corrected.*

Supplementary information, around the end of p. 1: can we say "concentration of radioactivity"?

*Yes, e.g. Bq/ml.*

---

## Author Response (AR2)

**Response to comments by the editor dated 17.11.2017**

P 1 L 19: Can you be more specific about what the implications are?

*We changed the sentence accordingly.*

P 2 L 7: Please adjust wording. There is no need for a soil to provide the P a plant needs. A plant has to cope with whatever it gets from the soil.

*We changed the sentence accordingly.*

P2 L 25 (or at a similarily suitable place). Please define the relationship between PBC and Km, possibly by introducing a new equation giving Km = concentration / exchange rate (or whatever would be the most appropriate description when comparing this to eq 1. You may well combine this with current equation 2.

*We see the editor's point that it would be nice to define the relationship between PBC and Km at this point and introduce an equation parallel to Eq. 1 explaining Km in like terms. However, this is not so straightforward. Km is the mean rate of exchange between phosphate ions in solution and inorganic phosphate in soil. This equates to the flux of P between the soil solution and the solid phase (mg P kg-1 min-1) divided by the soil solution concentration (mg P kg-1). However, it is not possible to measure this flux, rather Km can be used to calculate the flux. Thus we think including this definition might just confuse the reader.*

*We propose to change the text at L 25 to introduce and define Km rather than including another equation.*

P4 L 6: are you really referring to eq 2 (not 3)? Please carefully check all text references to equations.

*Thank you for catching this error. We corrected Eq. references here and elsewhere.*

P8 L 2: "On which time frame" ?

*We changed the wording to "on which time frame".*

P9 L 3-4: Given that Km is derived from n and m, Km is also only a statistical variable. Please check the wording here. I assume you want to say that your derived Km can be readily interpreted in terms of real-soil processes?

*We changed the sentence accordingly.*

P9 L 25: what kind of P is meant here?

*We added "availability" to make it clear that we mean P availability.*

[revised manuscript text omitted]